# Paternal indifference and neglect in early life and creativity: Exploring the moderating role of *TPH1* genotype and offspring gender

Qi Yu, Si Si, Shun Zhang, Jinghuan Zhang*

Department of Psychology, Shandong Normal University, Jinan, China

* zhangjinghuan@126.com

## Abstract

For further understanding the joint contribution of environment, heredity and gender to creativity, the present research examined the prospective impact of paternal indifference & neglect in early life, *TPH1* rs623580, offspring gender, and the interaction effects thereof on creativity in five hundred and thirty-nine unrelated healthy Chinese undergraduate students. Paternal indifference & neglect in early life was assessed on the Parental Bonding Instrument (PBI) and creativity on the Runco Creativity Assessment Battery (rCAB). Two primary findings emerged. Firstly, significant paternal indifference & neglect × *TPH1* genotype interaction effects were identified in predicting all three dimensions of creativity (fluency, originality, and flexibility). Paternal indifference & neglect in early life negatively predicted fluency, originality, and flexibility when individuals carry A allele of *TPH1* (rs623580). Secondly, there was a significant interaction effect of *TPH1* genotype by offspring gender on flexibility. Only in males, individuals who carry A allele were linked with lower level of flexibility compared to TT homozygote individuals. No significant three-way interaction was found. In conclusion, the current findings provided the first preliminary evidence for the moderation effect of *TPH1* on the relationship between parenting and creativity, and *TPH1*- offspring gender interaction on creativity; future studies are needed to validate these findings.

## Introduction

Creativity is defined as the capacity for producing something that is both novel and useful [1–3]. There is a consensus in the field that creativity involves in the improvement of technology, science, art, philosophy, or even all walks of life [4]. Previous studies have indicated that creativity is the major driving forces behind the progress of civilization [5, 6].

How biological and environmental factors foster or inhibit creativity has long been a subject of great interest for psychologists [7, 8]. For the biological factors, recent advances in molecular genetics have permitted psychologists to explore the underlying genetic basis of creativity, and several genes (e.g. *THP1*, *TPH2*) have been reported to associate with creativity [9–11]. For the environmental factors, parenting has been one of the most frequently investigated topics due to its crucial role in creativity [12, 13]. However, results from twin and adoption studies

**Data Availability Statement:** We have uploaded the minimal anonymized data set as Supporting Information files (S1_DATA).

**Funding:** This research was supported by National Natural Science Foundation of China (31470999,

31771235), Key special project of national key research and development program of China (SQ2017YFB1400102) and Shandong Provincial Institute of Qilu Cultural Studies to JZ. The funders had no role in study design, data collection and analysis, decision to publish, or preparation of the manuscript.

**Competing interests:** The authors have declared that no competing interests exist.

have indicated that creativity cannot be explained exactly by gene or environment alone [14, 15]. A growing evidence has highlighted the importance of Gene × Environment (G × E) interactions, in which the relationship between environmental factors (e.g. parenting) and child outcomes (e.g. antisocial behaviors, cognitive abilities, social function, wellbeing) might be moderated by genetic factors [16, 17]. Therefore, the primary purpose of present study was to explore the interaction effect of genetic and environmental factors on creativity.

Besides, previous study indicated that gender difference may contribute to the interaction effect of genetic and environmental factors on creativity [18]. Therefore, offspring gender was another variable recruited in this study. The objective of present study was to explore the joint contribution of environment, gene and offspring gender to creativity.

## Parental indifference & neglect and creativity

The early life family environment has long been recognized to influence creativity, among which parenting have received the most attention [19–21]. Parental indifference & neglect is a significant risk factor for children across their psychological and behavioral development and is usually linked with a variety of serious negative outcomes in adulthood [22–24], including psychological maladjustment, internalizing/externalizing behaviors, and negative personality dispositions of children [22, 25, 26].

According to Parental Acceptance-rejection Theory (PART), parental indifference refers to a mood state of parents distinguished by a lack of care, concern and interest of their children; while parental neglect refers to a behavioral response that parents fail to attend the physical, psychological, and social needs of their children appropriately [25, 27]. Although there existed difference between indifference and neglect in parenting behavior and affection, both indifferent and neglecting parents remain unavailable and unresponsive to their children's need, and it induces children to feel like they don't deserve to be loved and cared for [6], and makes the children too concerned with their own value without the energy to promote cognitive and emotional development. Recent empirical studies have indicated that parental indifference & neglect in early life negatively predict positive outcomes, such as cognition and intelligence [28–31]. Using the Audio-Computer Assisted Self Report Interview (ACASI), one study investigated the relation between multidimensional neglect and cognition, the result showed that children suffering neglect had lower overall cognitive performance in comparison with normative data [30]. Coincidentally, using the Wechsler Intelligence Scale for Children -Revised, Split-Half Short Form (WISC-R:SH), Kaufman et al. (1994) reported a direct relation of neglect to intelligence quotient (IQ), children who experienced the most severe parental neglect had the lowest IQ scale scores [31]. A further study demonstrated that the neglected children showed lower general intelligence and poorer executive decision than the controls [28]. Creativity and divergent thinking are deemed to be facets of intelligence in some intelligence models [9, 32, 33]. Based on the notion, parental indifference & neglect in early life may play a negative role in creativity in youth.

However, existing parenting research has documented that parental indifference & neglect in early life is not always deleterious. Previous studies provided evidence that parental indifference & neglect may positively relate to child's creativity. Albert (1992) reported that many geniuses and great eminences were suffered from parental indifference & rejection and poverty in early family environment [34]. Similarly, a longitudinal study, which aimed to reveal the association between parent-child relationships and creative personality traits, suggested that individuals with creative personality traits, such as self-sufficient, reserved, serious, adventurous, and sensitivity, were inclined to report their parents expressed more neglect and reject during the period of their growing up [35]. Inconsistent findings suggest that the relation

between parental indifference & neglect in early life and child developmental outcomes may be moderated by additional variables.

## Role of paternal indifference & neglect

Most studies in this research area focused on the effect of both parents' indifference & neglect to their children [25, 36]. Some researchers have noted that, fathers and mothers behave in a similar manner, whereas they play significant and differentiable role in the development of their children [37]. Mother's specific role is to provide a feeling of security, while the father's specific role is to prompt the children to attain higher levels of success [38]. An ever-expanding line of research indicated that fathers played an important role in children's psychological and behavioral development, including academic achievement, cognitive development, behavioral or emotional regulation and so forth [39, 40]. A study of American fathers of 2-year-old children with low socioeconomic strata reported that fathers with more responsive/didactic behavior (including responsiveness, emotional regulations, and communication) toward their children were nearly five times more likely to have children within the normal developmental range including problem solving and memory than other fathers [41]. Given that problem solving has long been viewed as a characteristic of creative activity [42], and information processing mechanisms underlying creativity has been suggested in relation to various aspects of memory [43]. It is reasonable to assume that paternal indifference & neglect in early life may play an important role in creativity. Thus, the present study was designed to investigate the particular relation of paternal indifference & neglect in early life to creativity in youth.

## *TPH1* rs623580 and creativity

Studies utilizing behavior genetic research designs have demonstrated both genetic and environmental factors have influence on individual's creativity [44]. Recent advances in molecular genetic studies have permitted direct exploring the underlying mechanism of the G × E interaction via identifying specific genes or locus associated with creativity. Empirical research showed a genetic variant in the dopamine D2 receptor gene *(DRD2)*, rs1799732 polymorphisms, moderated the relation between authoritarian parenting and creativity [3]. Thus, we postulated in this line that the relation between paternal indifference & neglect in early life and creativity in youth may be moderated by genetic variants.

Several lines of research have indicated that the *TPH1* genotypes involve in creativity. Using inventiveness battery of the Berlin Intelligence Structure Test (BIS), Reuter et al. (2006) reported that *TPH1* rs1799913 (A779C) polymorphism was significantly associated with creativity. Similar findings, using Divergent Thinking Test (DT Test), indicated that *TPH1* rs1799913 polymorphism was significantly associated with ideational fluency [10]. To further elucidate the role of *TPH1* in creativity, by including both related functional SNPs and tag SNPs, a recent study comprehensively explored the correlation between *TPH1* genetic variants and creative potential measured by DT Test [11]. Although failed to replicate the correlation of *TPH1* rs1799913 and creativity, the study reported a new *TPH1* genetic variate, rs623580 (T3804A), associated with both verbal and figural fluency.

*TPH1* rs623580 located in the exon 1c & intron1 within the 5'- UTR of the *TPH1* gene at human chromosome 11 [45]. *TPH1* is the rate limiting enzyme in the biosynthesis pathway of the neurotransmitter 5-hydroxytryptamine (5-HT, Serotonin) and therefore a critical step in 5-HT functioning [46]. *TPH1* gene expression is limited to a few specialized tissues, including brainstem raphe neurons, pinealocytes, the central nervous system (CNS), and part of the peripheral serotonergic nervous system [47]. Using a GWAS of 909 families (three members per family including ADHD patients and their parents), Sonuga-Barke et al. (2008) reported

nominal evidence for interaction between *TPH1* rs623580 and parental criticism when predicting conduct disorder symptom [48]. Although the underlying mechanism was still unclear, this study provided the primary evidence for *TPH1* rs623580 moderate the relation between adverse environments and outcomes. Therefore, the present study designed to test whether the relation between paternal indifference & neglect in early life and creativity in youth was moderate by *TPH1* rs623580.

### Role of offspring gender

Beside genetic variants, there exists growing evidence that the role of paternal indifference & neglect in offspring developmental outcomes may be different for boys and girls. Several studies suggest that father is the most significant model for boys' identification [49], if the father is unavailable, then the boys have a greater likelihood of engaging in the negative outcomes [50, 51]. Other study, however, showed that women were more likely than men to be influenced by paternal indifference & neglect. Using clinical and non-clinical subjects, Handa et al. [52] reported that in female patients, low paternal care in early life was significantly associated with a higher likelihood of showing symptoms of prolonged depression, while in male patients, no correlation between low paternal care in early life and prolonged depression was found. Thus, in the present study we hypothesized that offspring gender may moderate the relationships between paternal indifference & neglect in early life and creativity in youth.

Moreover, previous research tested the relation between *TPH1* rs623580 and Depressive Disorder in Chinese subjects, the result showed that, in women the frequencies of the genotypes and alleles of *TPH1* rs623580 (A allele) in depressive disorder group were statistically different from those in normal control group, but not in men [53]. Although the underlying mechanism of the gender difference was not clear, this study suggested that the relation between *TPH1* rs623580 and depressive disorder might be different between women and men. Thus, we also postulated in this line that the relation between *TPH1* rs623580 and creativity might be moderated by offspring gender.

Although lacking of the empirical evidence, it has been suggested in the literature that the difference in creativity may be as a result of a combination of environmental, genetic, and gender. Abra and Valentine-French (1991) considered that gender differences in creative achievement depends on both biological and environmental factors. They highlighted that the effect of possible genetic and environmental sources of such differences should be noted. Because males and females differ in both factors, either or both may lead to the differences in creative achievement [18].

Based on this review of the literature, the current study aimed to explore the impact of paternal indifference & neglect in early life, *TPH1* rs623580, offspring gender, and the interaction effects thereof on creativity in youth. We postulated that paternal indifference & neglect in early life would negatively predict creativity in youth. We also assumed that *TPH1* rs623580 polymorphism and offspring gender would moderate the influence of paternal indifference & neglect in early life on creativity in youth.

## Methods

### Participants and procedure

Participants included 539 (183 males and 356 females, gender was determined by self-report) unrelated healthy Han Chinese undergraduate students with an average age of 18.93 years (SD = 1.084, range = 17–22) from Shandong Normal University. None of the participants had been hospitalized for head trauma, psychiatric or neurologic reasons and none abused alcohol or drugs. The study protocol was approved by the Institutional Review Board of Shandong

Normal University. Written informed consent for genetic analysis was obtained from each participant after a description and explanation of the study.

*TPH1* **rs623580.** DNA was extracted from peripheral venous blood samples using the QIAamp DNA Mini Kit (Qiagen, Valencia, CA, USA). Genotyping was carried out by a technician blind to other data from the research project. The single nucleotide polymorphisms (SNPs) were genotyped at the Beijing Genomics Institute-Shenzhen (BGI-Shenzhen, Shenzhen, China) using the Sequenom®MassARRAY®iPLEX system (Sequenom, San Diego, CA, USA). A customized set of SNPs was provided to BGI-Shenzhen by the investigator and BGI-Shenzhen provided the final oligonucleotides sequences to be used. Reverse and extension primers were designed using the MassARRAY Assay Design 3.0. For quality control, 5% random DNA samples were re-genotyped for each SNP, yielding a reproducibility of 100%. The *TPH1* rs623580 polymorphism was assessed as part of the SNP panel and met the criteria mentioned above. Genotype distribution of *TPH1* rs623580 for AA was 14.5% (n = 78), AT was 50.2% (n = 271), and TT was 35.3% (n = 190). Consistent with previous research [54], AA and AT genotypes were combined and compared with the TT group. Allelic frequency of *TPH1* rs623580 is presented in Table 1.

## Measures

**Creative potential measures.** Creativity was measured by Figural Divergent Thinking Test (Figural DT Test) selected from the Runco Creativity Assessment Battery (rCAB; Creativity Testing Service, Bishop, GA, USA). The Figural DT Test includes three items. A line-drawings was represented in each item, and participants were asked to list as many responses as they can in four minutes. According to the guideline of Creativity Testing Service, the following three scores were obtained: fluency, flexibility, and originality [55]. Fluency score was obtained by counting the number of unduplicated ideas provided by each participant. Originality score was calculated by counting the number of unusual ideas provided by each participant. Unusual ideas were defined as ideas given by less the 5% of the respondents in the sample. To score flexibility, a category list was first generated for each item based on the responses provided by all respondents. The category list was generated from each set of answers via the categorizing of responses which have common characteristics (e.g., "cake" and "noodle" were classified as "food", "hook" and "bench" were classified as "furniture", "bullet" and "arrow" were classified as "weapon", etc.). Flexibility score was computed by counting the number of different categories used in one participant's responses [3, 11, 56, 57]. Two trained raters (both were psychology graduate students from Shandong Normal University) were engaged to score all those ideas. The Chinese version of this measure was a widely used noninvasive measure and demonstrated adequate reliability and validity [3, 11, 20, 56, 57]. The inter-rater reliabilities for all the three scores in the present study were higher than .95; therefore, the final scores were obtained by averaging scores from the two raters. In current study, the Cronbach's alpha was .86 for fluency, .69 for flexibility, and .83 for originality.

**Parental Bonding Instrument (PBI).** The Parental Bonding Instrument is a 25-item self-rating questionnaire designed to measure the quality of the attachment or bond between parents and their children, based on the memory of participants regarding their parents before

**Table 1. Frequency of the *TPH1* rs623580.**

| *TPH1* rs623580 | Full sample |
|---|---|
| 1 | 349(64.7%) |
| 0 | 190(35.3%) |

Frequency of each allele (0 = TT, 1 = AA & AT) and the corresponding percentage (in parentheses) are reported.

their age of 16 [58]. Six items define the "care", in which the higher the score, the higher the affection and warmth exercised by their parents; six items define the "indifference & neglect", in which the higher the score, the higher the rejection and neglect exercised by their parents; seven items establish the "overprotection", in which the higher the score, the higher the over involvement attitude and psychological control from their parents; six items on the "autonomy", in which the higher the score, the higher the encouragement of independence attitude and psychological autonomy from parents [59]. Participants scored each of their parents separately, on a 4-point Likert-type scale ranging from 0 ("very unlike") to 3 ("very like"). The Chinese version of this measure was available and established reliability and validity [60]. In this study, the Cronbach's alpha coefficients of four subscales were .84 for care, .78 for indifference & neglect, .82 for overprotection, and .78 for autonomy.

### Data analysis

To test whether the relationships between paternal indifference & neglect and creativity (fluency, flexibility, originality) were moderated by *TPH1* rs623580 and offspring gender, a series of hierarchical regression analyses were performed with mean centered variables. Paternal care was significantly related to paternal indifference & neglect and was therefore included in the regressions. Age and paternal care were included as covariates in the first regression step. In the second step, creativity (fluency, flexibility, originality) was predicted from the main effects of offspring gender (male coded as 1 and female as 0), paternal indifference & neglect, and *TPH1* rs623580. Then the moderator terms (the interaction between paternal indifference & neglect, *TPH1* rs623580, and offspring gender) was added in the third step.

Because all three-way interaction effect on three outcomes were not significant, we performed two two-way interaction separately on each outcome. When significant paternal indifference & neglect × *TPH1* rs623580 and *TPH1* rs623580 × offspring gender interactions were found, the nature of the interactions was tested by post-hoc analyses. The SPSS version 16.0 was used for analysis.

## Results

Table 2 reports the correlations, means, and standard deviations of the variables of this study. Paternal care was positively correlated with fluency ($r = 0.127$, $p<0.01$), flexibility ($r = 0.112$, $p<0.01$), and originality ($r = 0.117$, $p<0.01$). Paternal indifference & neglect were negatively correlated with fluency ($r = -0.107$, $p<0.05$), flexibility ($r = -0.085$, $p<0.05$), and originality ($r = -0.089$, $p<0.05$). There were evidences for gender differences in fluency ($r = -0.278$, $p<0.01$), flexibility ($r = -0.225$, $p<0.01$), and originality ($r = -0.195$, $p<0.01$), but not in *TPH1* rs623580 ($r = -0.061$, $p>0.05$) and each of those paternal bonding variables ($ps>0.05$). *TPH1* rs623580 was not correlated with any paternal bonding variables, offspring gender, and each of the outcome variables ($ps>0.05$). The findings of the interaction effect of paternal indifference & neglect and *TPH1* rs623580 on the outcome variables are summarized in Table 3. The findings of the interaction effect of *TPH1* rs623580 and offspring gender on the outcome variables are summarized in Table 4.

### Paternal indifference & neglect and fluency: *TPH1* rs623580 and offspring gender as moderators

Results showed that both paternal indifference & neglect and offspring gender had direct main effects on fluency ($B = 1.577$, $p<0.05$; $B = -1.936$, $p<0.01$), while the main effect for *TPH1* rs623580 was not significant (AA & AT = 1, $B = -0.351$, $p = 0.437$). The three-way interaction of paternal indifference & neglect, *TPH1* rs623580, and offspring gender on fluency was not

**Table 2. Correlations among primary study variables.**

| Variable | 1 | 2 | 3 | 4 | 5 | 6 | 7 | 8 |
|---|---|---|---|---|---|---|---|---|
| 1.age | — | | | | | | | |
| 2. offspring gender | .101* | — | | | | | | |
| 3.rs623580 | -.027 | -.061 | — | | | | | |
| 4. PC | -.102* | -.045 | -.019 | (.84) | | | | |
| 5. PI | .077 | .075 | -.006 | -.741** | (.78) | | | |
| 6. fluency | -.012 | -.278** | -.058 | .127** | -.107* | (.86) | | |
| 7. originality | -.004 | -.195** | -.050 | .117** | -.089* | .930** | (.83) | |
| 8. flexibility | -.031 | -.225** | -.073 | .112** | -.085* | .819** | .741** | (.69) |
| Mean | 18.91 | .34 | .65 | 2.03 | .76 | 10.05 | 4.88 | 5.10 |
| SD | 1.08 | .47 | .48 | .59 | .54 | 4.20 | 3.00 | 1.26 |

Male = 1, Female = 0; PC = Paternal care, PI = Paternal indifference & neglect; the Cronbach's alpha coefficients of PC, PI and fluency, originality, flexibility were reported in the parentheses;

*$p < .05$,

**$p < .01$.

significant ($B = 0.371$, $p = 0.788$), but there was a significant two-way interaction of paternal indifference & neglect and *TPH1* rs623580 ($B = -0.193$, $p<0.05$). This two-way interaction remained significant after the non-significant three-way and all non-significant two-way interaction terms were dropped and a reduced model was run ($B = -0.182$, $p<0.05$) (see Table 3).

The significant interaction term of paternal indifference & neglect and *TPH1* rs623580 on fluency was tested for each *TPH1* genotype group. Results of the regression for AA / AT genotypes indicated that paternal indifference & neglect was related to lower level of fluency ($B = -1.429$, $p<0.05$, 95% $CI = -2.240$ to $-0.617$). In contrast, results of the regression for TT genotype indicated that paternal indifference & neglect was not associated with fluency ($B = 0.310$, $p>0.05$, 95% $CI = -0.787$ to $1.407$). Regression lines depicting levels of paternal indifference & neglect for AA / AT genotypes and TT genotype are plotted in Fig 1A.

**Paternal indifference & neglect and originality: *TPH1* rs623580 and offspring gender as moderators.** Results showed that both paternal indifference & neglect and offspring gender

**Table 3. Hierarchical linear regression analysis testing the effects of paternal indifference & neglect, TPH genotype and their interaction on creativity.**

| Variables | Model 1 | Model 2 | Model 3 | Model 4 | Model 5 | Model 6 | Model 7 | Model 8 | Model 9 |
|---|---|---|---|---|---|---|---|---|---|
| | Fluency | Fluency | Fluency | Originality | Originality | Originality | Flexibility | Flexibility | Flexibility |
| | β | β | β | β | β | β | β | β | β |
| Age | .001 | .000 | -.010 | .008 | .006 | -.002 | -.020 | -.022 | -.031 |
| PC | .125** | .104 | .105 | .117** | .112 | .112 | .108* | .104 | .105 |
| PI | | -.028 | .119 | | -.005 | .132 | | -.004 | .138 |
| rs623580 | | -.056 | -.056 | | -.047 | -.048 | | -.071 | -.071 |
| PI × rs623580 | | | -.182* | | | -.170* | | | -.175* |
| F | 4.275* | 2.593* | 3.375** | 3.650* | 2.127 | 2.826* | 3.401* | 2.386* | 3.103** |
| R² | .016 | .019 | .031 | .013 | .016 | .026 | .013 | .018 | .028 |
| △R² | .012 | .012 | .022 | .010 | .008 | .017 | .009 | .010 | .019 |

Male = 1, Female = 0; PC = Paternal care, PI = Paternal indifference & neglect;

*$p < .05$,

**$p < .01$.

**Table 4. Hierarchical linear regression analysis testing the effects of TPH genotype, offspring gender and their interaction on creativity.**

| Variables | Model 1 | Model 2 | Model 3 | Model 4 | Model 5 | Model 6 | Model 7 | Model 8 | Model 9 |
|---|---|---|---|---|---|---|---|---|---|
| | Fluency | Fluency | Fluency | Originality | Originality | Originality | Flexibility | Flexibility | Flexibility |
| | β | β | β | β | β | β | β | β | β |
| Age | -.012 | .015 | .011 | -.004 | .014 | .011 | -.031 | -.011 | -.017 |
| rs623580 | | -.074 | -.039 | | -.062 | -.028 | | -.087* | -.018 |
| offspring gender | | -.283*** | -.220** | | -.200*** | -.141* | | -.228*** | -.106 |
| rs623580×offspring gender | | | -.085 | | | -.080 | | | -.165* |
| F | .078 | 15.994*** | 12.337*** | .009 | 7.796*** | 6.131*** | .524 | 10.875*** | 9.451*** |
| R$^2$ | .000 | .082 | .085 | .000 | .042 | .044 | .001 | .058 | .066 |
| ΔR$^2$ | -.002 | .077 | .078 | -.002 | .037 | .037 | .000 | .052 | .059 |

Male = 1, Female = 0; PC = Paternal care, PI = Paternal indifference & neglect;

*$p < .05$,

**$p < .01$,

***$p < .001$.

had direct main effects on originality ($B = 1.253$, $p<0.05$; $B = -0.876$, $p<0.05$), while the main effect for *TPH1* rs623580 was not significant (AA & AT = 1, $B = -0.181$, $p = 0.583$). Although the three-way interaction of paternal indifference & neglect, *TPH1* rs623580, and offspring gender on originality was not significant ($B = 0.402$, $p = 0.689$), there was a significant two-way interaction of paternal indifference & neglect and *TPH1* rs623580 ($B = -0.190$, $p<0.05$). This two-way interaction remained significant after the non-significant three-way and all non-significant two-way interaction terms were dropped and a reduced model was run ($B = -0.170$, $p<0.05$) (see Table 3).

The significant interaction term of paternal indifference & neglect and *TPH1* rs623580 on originality was tested for each *TPH1* genotype group. Results of the regression for AA / AT genotypes indicated that paternal indifference & neglect was related to lower level of originality ($B = -0.892$, $p<0.05$, 95% $CI = -1.457$ to $-0.326$). In contrast, results of the regression for TT genotype indicated that paternal indifference & neglect was not associated with originality ($B = 0.269$, $p>0.05$, 95% $CI = -0.558$ to $1.096$). Regression lines depicting levels of paternal indifference & neglect for AA / AT genotypes and TT genotype are plotted in Fig 1B.

**Paternal indifference & neglect and flexibility: *TPH1* rs623580 and offspring gender as moderators.** Results revealed no significant main effects of paternal indifference & neglect ($B = 0.445$, $p = 0.067$), *TPH1* rs623580 ($B = -0.050$, $p = 0.718$), and offspring gender ($B = -0.283$, $p = 0.124$). The three-way interaction of paternal indifference & neglect, *TPH1* rs623580, and offspring gender on flexibility was not significant ($B = 0.205$, $p = 0.625$). However, two significant two-way interactions emerged.

First, there was a significant interaction of paternal indifference & neglect and *TPH1* rs623580 ($B = -0.193$, $p<0.05$). This two-way interaction remained significant after the non-significant three-way and all non-significant two-way interaction terms were dropped and a reduced model was run ($B = -0.175$, $p<0.05$) (see Table 3). The significant interaction term of paternal indifference & neglect and *TPH1* rs623580 on flexibility was tested for each *TPH1* genotype group. Results of the regression for AA / AT genotypes indicated that paternal indifference & neglect was related to lower level of flexibility ($B = -0.369$, $p<0.05$, 95% $CI = -0.610$ to $-0.128$). In contrast, results of the regression for TT genotype indicated that paternal indifference & neglect was not associated with flexibility ($B = 0.13$, $p>0.05$, 95% $CI$

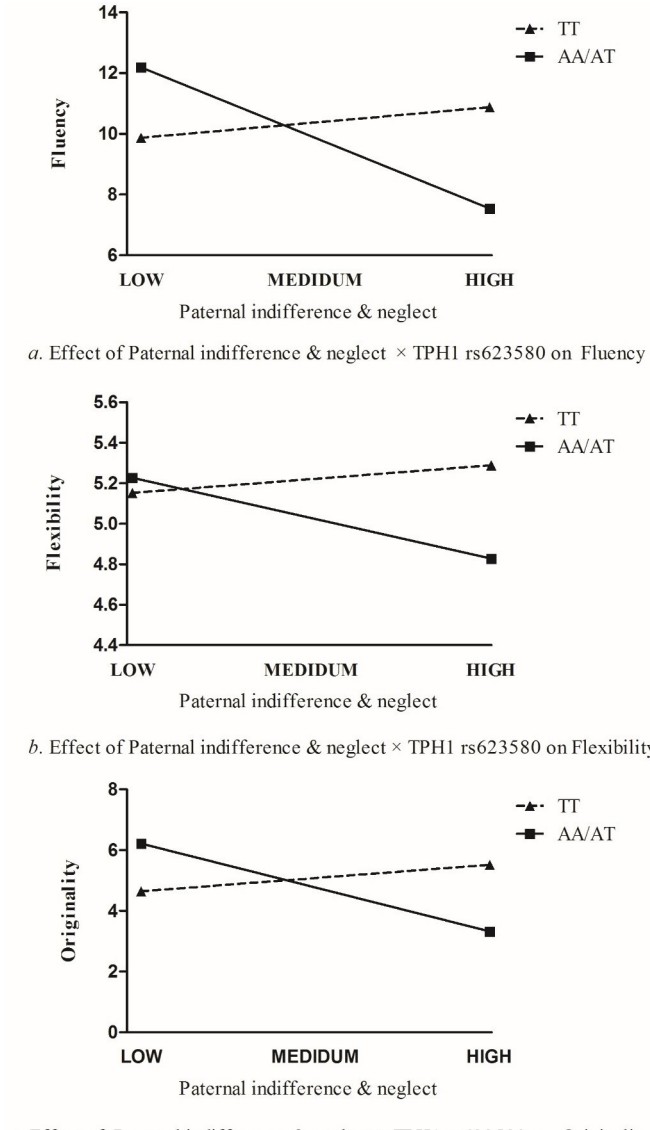

*a.* Effect of Paternal indifference & neglect × TPH1 rs623580 on Fluency

*b.* Effect of Paternal indifference & neglect × TPH1 rs623580 on Flexibility

*c.* Effect of Paternal indifference & neglect × TPH1 rs623580 on Originality

**Fig 1. Effect of paternal indifference× *TPH1* rs623580 on fluency, flexibility, and originality.**

= -0.211 to 0.464). Regression lines depicting levels of paternal indifference & neglect for AA / AT genotypes and TT genotype are plotted in Fig 1C.

Second, an interaction emerged between *TPH1* rs623580 and offspring gender ($B$ = -0.159, $p < 0.05$). This two-way interaction remained significant after the non-significant three-way and all non-significant two-way interaction terms were dropped and a reduced model was run ($B$ = -0.165, $p < 0.05$) (see Table 4). The significant interaction term of *TPH1* rs623580 and off-spring gender on flexibility was tested for each *TPH1* genotype group. Results of the regression for AA / AT genotypes indicated that male was related to lower level of flexibility ($B$ = -0.801 $p < 0.001$, 95% $CI$ = -1.073 to -0.529). In contrast, results of the regression for TT genotype indicated that offspring gender was not associated with flexibility ($B$ = -0.291, $p > 0.05$, 95% $CI$ = -0.660 to 0.078). Regression lines depicting levels of offspring gender for AA / AT genotypes and TT genotype are plotted in Fig 2.

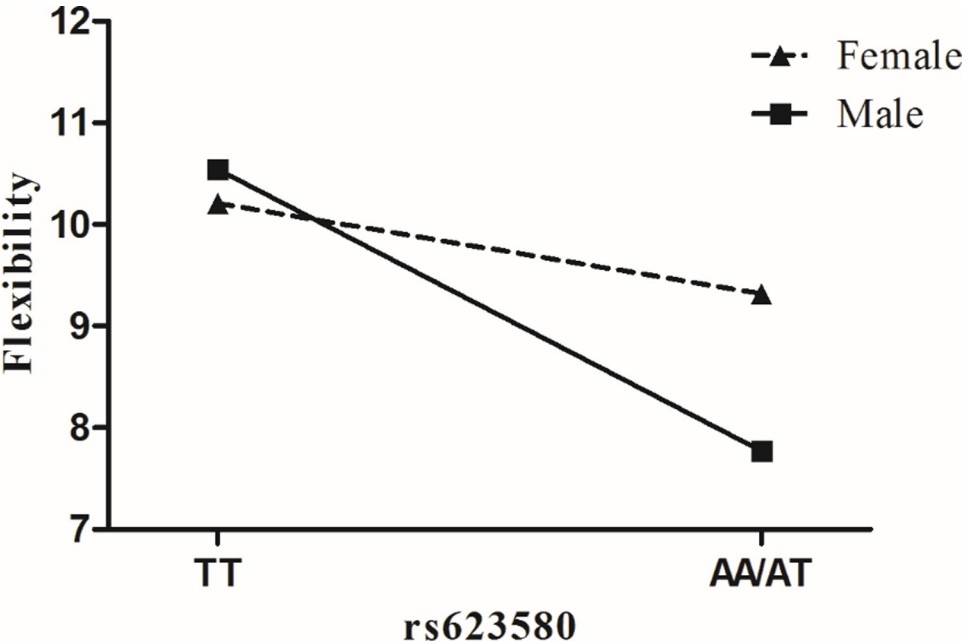

**Fig 2. Effect of *TPH1* rs623580 ×offspring gender on flexibility.**

## Discussion

This study aimed to examine the impact of paternal indifference & neglect in early life, *TPH1* rs623580, offspring gender, and the interaction effects thereof on creativity in youth. Two primary findings emerged. First, paternal indifference & neglect in early life negatively predicted all three dimensions of creativity in youth (fluency, flexibility and originality) when individuals carry A allele of *TPH1* rs623580. Second, males who carry A allele of *TPH1* rs623580 were linked with lower level of flexibility compared to TT homozygote carriers.

Firstly, present study provided evidence for paternal indifference & neglect in early life negatively predicted on creativity in youth (fluency and originality). These findings were consistent with previous research in which a negative relation between paternal rejection and adolescents' creativity was demonstrated in Chinese samples [61]. Given that indifferent and neglecting father usually remains psychologically and physically unresponsive or even inaccessible, they may be prejudicial to child's psychological security [25]. Psychological security has been demonstrated to positively predict creativity [62, 63]. Therefore, it is reasonable to speculate that paternal indifference & neglect in early life may be adverse to individual's psychological security, which negatively impacts on creativity in youth.

These findings of the direct effects of paternal indifference & neglect in early life on creativity in youth were incongruent with prior studies in Western settings [34, 35]. In contrast with Westernized cultures, Chinese culture is widely characterized as collectivistic, which emphasize interpersonal relatedness [64, 65]. Children may be more sensitive to paternal indifference/neglect in Chinese societies than in Western societies [25, 66]. However, it is difficult to compare the correlations for the two cultural groups due to lack of data on the relations between paternal indifference & neglect and creativity in Western studies. Further examination of this issue is needed in future cross-cultural research.

Second, consistent with our expectation, the relation between paternal indifference & neglect in early life and creativity in youth was moderate by *TPH1* rs623580. The negative

influence of paternal indifference & neglect in early life on creativity in youth was only present in individuals who carry A allele of *TPH1* rs623580 but not the carriers of the TT genotype. This finding suggested that carrying the A allele of *TPH1* rs623580 may increase the vulnerability to paternal indifference & neglect in early life, and pose a risk for creativity in youth. Paternal indifference & neglect in early life has been identified as a potent source of stress, and has been suggested to have a pervasive influence on children's psychological and biological regulatory processes [67]. Molecular genetics research has demonstrated that *TPH1* mRNA expresses in the hypothalamus and the neuronal *TPH1* protein expresses in the anterior pituitary. These findings suggested that *TPH1* may involve in hypothalamic-pituitary-adrenal (HPA) axis regulation and may influence on stress-response mechanisms in the brain [68, 69, 70]. Although *TPH1* rs623580 does not result in an amino acid substitution as located in a regulatory region, it may affects in *TPH1* enzyme activity [48]. Therefore, it is possible that *TPH1* rs623580 may moderate the negative relation between paternal indifference & neglect in early life and creativity in youth via regulating the stress-response processes. Specifically, compared with the TT homozygote individuals, the A allele carriers may have less capacity to cope with the stress due to paternal indifference & neglect in early life, and to withstand the corrosive effect of it effectively, which in turn lead them to the damaging consequences [71, 72].

Third, consistent with our expectation, the relation between *TPH1* rs623580 and creativity was moderated by offspring gender. Specifically, males who carrying the A allele showed lower flexibility than the TT carriers. This result suggested that A allele of *TPH1* rs623580 may be a risk allele for decreasing creativity, at least in males. Animal research has indicated that sex hormones, including estrogen and progesterone, can increase *TPH1* expression in the central nervous system of primates [73]. It could be speculated that the gender difference in the relation of *TPH1* rs623580 A allele to flexibility might be partly due to sex hormones regulation, that is lower level of estrogen and progesterone in male may down-regulate expression of *TPH1*. Although the underlying mechanism of the interaction effect is not yet clear, the result suggested that *TPH1* rs623580 may involve in gender difference in creativity.

Fourth, inconsistent with our speculation, the three-way interaction was not significant, suggesting that the relation of paternal indifference & neglect in early life and *TPH1* rs623580 to creativity is the same for both males and females. This result suggested that *TPH1* rs623580, but not gender, may be a crucial factor helped to explain those inconsistent findings on the relation between paternal indifference & neglect and creativity in previous research. This finding emphasized that father involvement plays an important role in the development of creativity for both boys and girls, especially for the children with the A allele of *TPH1* rs623580. Fathers should take more time to engage directly with their children in their early lives.

Several limitations of this study should be addressed. Firstly, the present study employed a retrospective design to explore the influence of paternal indifference & neglect in early life on creativity. Longitudinal study from early childhood to young adulthood is needed to understand the dynamic association between early life family environment and creativity. Secondly, the assessment of early life paternal indifference & neglect in the present study was limited in self-report measure, which may only reflect perceived paternal indifference & neglect of participants, not objectively observed paternal indifference & neglect. Future study simultaneously including the parents and observer reports of early life family environment would provide more convincing results. Third, the present study used a relatively homogenous sample consisting of Chinese undergraduate students. As the genetic backgrounds vary for different ethnic populations, the generalization of the present findings to other samples is limited. Future research across populations of different genetic and cultural backgrounds are warranted to examine what extent the present findings can be generalized to other samples.

These limitations notwithstanding, some valuable information can be derived from our findings. Drawing upon gene × environment and gene × gender interaction research, this study provided evidence that carrying A allele of *TPH1* rs623580 may be a significant risk factor of creativity decline. The findings of present study contribute to a better understanding of the role of genetic factors in the relationship between parenting and creativity. In addition, our findings may also provide a new perspective to reevaluate the genetic basis of gender difference in creativity.

## Supporting information

**S1 Data. Data underlying the presented results.**
(SAV)

## Acknowledgments

All authors wish to acknowledge and thank Dr. Mark A. Runco (Torrance Creativity Center, University of Georgia) for the directions and help about the Divergent Thinking Test scoring.

## Author Contributions

**Methodology:** Qi Yu, Si Si.

**Supervision:** Jinghuan Zhang.

**Writing – original draft:** Qi Yu.

**Writing – review & editing:** Si Si, Shun Zhang, Jinghuan Zhang.

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
