## [Decision Letter · Decision Letter 0]

3 Dec 2019

PONE-D-19-21789

Paternal Indifference and neglect in early life and Creativity: Exploring the Moderating Role of TPH1 genotype and Offspring's Gender

PLOS ONE

Dear Dr Zhang,

Thank you for submitting your manuscript to PLOS ONE. After careful consideration, we feel that it has merit but does not fully meet PLOS ONE’s publication criteria as it currently stands. Therefore, we invite you to submit a revised version of the manuscript that addresses the points raised during the review process.

We would appreciate receiving your revised manuscript by Jan 17 2020 11:59PM. To enhance the reproducibility of your results, we recommend that if applicable you deposit your laboratory protocols in protocols.io, where a protocol can be assigned its own identifier (DOI) such that it can be cited independently in the future. For instructions see: http://journals.plos.org/plosone/s/submission-guidelines#loc-laboratory-protocols

We look forward to receiving your revised manuscript.

Kind regards,

Dongtao Wei

Academic Editor

PLOS ONE

Journal Requirements:

2. Please outline how gender was determined in your Methods section.

Additional Editor Comments (if provided):

Reviewers' comments:

Reviewer's Responses to Questions

**Comments to the Author**

1. Is the manuscript technically sound, and do the data support the conclusions?

Reviewer #1: Yes

Reviewer #2: Yes

2. Has the statistical analysis been performed appropriately and rigorously? 

Reviewer #1: Yes

Reviewer #2: Yes

3. Have the authors made all data underlying the findings in their manuscript fully available?

Reviewer #1: Yes

Reviewer #2: No

4. Is the manuscript presented in an intelligible fashion and written in standard English?

Reviewer #1: Yes

Reviewer #2: Yes

5. Review Comments to the Author

Reviewer #1: This manuscript presents an evidence from a gene-environment interaction study on the impact of paternal indifference & neglect in early life, TPH1 rs623580, offspring's gender, and the interaction effects thereof on creativity. Basically, authors are well experienced in these sorts of studies and most of the methods taken are standard or within the accepted range. The sample size is another strength for this kind of study. Mostly, I have difficulty in finding major problems, but as for the following points need to be addressed in order to improve the manuscript.

1. The theoretical background of this study seems to be weak and unconvincing. Why, for instance, should parental indifference & neglect be associated with creativity? And the paper examined the TPH1 rs623580 × offspring’s gender interactions on creativity, but it seems lacked the theoretical or empirical evidence in introduction part.

2. The Parental Bonding Instrument consist of care, indifference & neglect, overprotection and autonomy, but why only care and indifference & neglect was used in the present paper. Were overprotection and autonomy related with creativity?

3.Whether mean centering was performed before moderation analysis?

4. Were scores of the creativity (fluency, flexibility and originality) normally distributed? If not, how was this accounted for in the statistical model that calls for normality?

Reviewer #2: Title: Paternal Indifference and neglect in early life and Creativity: Exploring the Moderating Role of TPH1 genotype and Offspring’s Gender

The manuscript “Paternal Indifference and neglect in early life and Creativity: Exploring the Moderating Role of TPH1 genotype and Offspring’s Gender” presented a meaningful study to explore the prospective impact of paternal indifference and neglect in early life, TPH1 rs623580, offspring’s gender, and their interaction effects on creativity. Drawing upon gene × environment and gene × gender interaction research, this paper has considerable strengths and may eventually be an important contribution to the literature. However, there are several significant issues remain to be considered:

Major concerns:

1. The overall structure of Introduction is slightly confusing and there is a lack of clarity. For example, the part subtitled “TPH1 rs623580 and creativity” in Introduction was actually about the relation between paternal indifference & neglect in early life and creativity in youth may be moderated by genetic variants.

2. It’s better going deeper into Discussion. It is not clear what these findings mean for our further understanding of the development of creativity as well as the cultivation of creativity. It should be discussed further.

3. Both “moderation” and “interaction effects” are very much similar to each other. Mathematically, they both can be modelled by using product term in the regression equation. Researchers often use the two terms as synonyms but there is a thin line between interaction and moderation. When we say X and Z interact in their effects on an outcome variable Y, and there is no real distinction between the role of X and the role of Z. They are both considered predictor variables. Then we identify this effect as interaction effect. While, in case we have clear distinction between the predictor and moderator variables (on the basis of theory) and we are interested to see the impact of predictor on response (affected by moderator), then this effect is known as moderation effect. The authors should carefully choose the term which is more suitable to answer their research question.

4. The Results reported the findings of the interaction effect of paternal indifference & neglect and TPH1 rs623580 as well as the interaction effect of TPH1 rs623580 and offspring’s gender on each outcome variable (fluency, flexibility, and originality), but the interaction effect of paternal indifference & neglect and TPH1 rs623580 on fluency, flexibility, and originality didn’t be introduced in detail in Abstract but stated simply “creativity”. Moreover, if there was a different pattern of results for each outcome variable (fluency, flexibility, and originality), it had better be discussed properly in Discussion.

Related to this is another issue: in METHODS, the authors did not introduce how they obtained creativity score, although they explained how they obtained fluency, flexibility, and originality scores.

5. Previous studies indicated that gender difference might be attributed to the interaction effect of genetic and environmental factors on creativity. What is the relationship between the findings in the present study and in previous studies? The cause of non-significant three-way interaction effect on three outcome variables was not discussed in Discussion.

Minor concerns：

1. The necessary information on Measures in METHODS should be included. For example, “Originality score was calculated by counting the number of unusual ideas given by less the 5% of the sample.” Please described in much more detail about “the sample”. Similarly, “To score flexibility, a category list was first generated for each item, and the flexibility score was the number of different categories used in one participant’s ideas.” Please described in much more detail about “a category list”.

2. It is noted that the manuscript needs careful editing. The overall language of the paper needs to be improved. Some sentences are awkward.

For example:

Syntax errors：“One possible explanation is that the influence of parental indifference & neglect to children may be differ between mother and father.”

“It is postulated that paternal indifference & neglect in early life would be negatively predict creativity in youth.”

“The findings of the present study contribute to further understanding the role of genetic factors in the pathways that how the early life family environment shapes creativity in adulthood.”

---

## [Author Response · Author response to Decision Letter 0]

6 Feb 2020

Responds to the reviewers’ comments: 

Reviewer #1: 

1. Response to comment: The theoretical background of this study seems to be weak and unconvincing. Why, for instance, should parental indifference & neglect be associated with creativity? And the paper examined the TPH1 rs623580 × offspring’s gender interactions on creativity, but it seems lacked the theoretical or empirical evidence in introduction part.

Reply: Thank you for your helpful advice. We have revised our introduction part to better illustrate the theoretical and empirical background of this study. We have added a paragraph subtitled “Role of paternal indifference & neglect” (Page 6, Line 94) to specify the reasons for our study of the relationship between parental indifference & neglect and creativity. Secondly, we have added three paragraphs to specify the reasons for our study examining the paternal indifference & neglect × offspring gender interaction(Page 8, Line 152), the TPH1 rs623580 × offspring gender interaction (Page 9, Line 165), and the three-way interaction(Page 9, Line 174).

2. Response to comment: The Parental Bonding Instrument consist of care, indifference & neglect, overprotection and autonomy, but why only care and indifference & neglect was used in the present paper. Were overprotection and autonomy related with creativity?

Reply: Thank you for your helpful advice. The reason why we only used care and indifference & neglect in this study is that the role of paternal indifference & neglect in creativity is what we are interested in. Previous research has reported that the two subscales parental care and indifference & neglect are highly related. Therefore, we also used care in the present paper. Although overprotection and autonomy have been reported to be relate to creativity, they are not the issues to be investigated in this study. Therefore, overprotection and autonomy were not used in the present paper.

3. Response to comment: Whether mean centering was performed before moderation analysis?

Reply: Thank you for your helpful advice. Mean centering was performed before moderation analysis. We have added the description of mean centering in our revised manuscript(Page 13, Line 257).

4. Response to comment: Were scores of the creativity (fluency, flexibility and originality) normally distributed? If not, how was this accounted for in the statistical model that calls for normality?

Reply: Thank you for your helpful advice. The scores of the creativity (fluency, flexibility and originality) were not normally distributed. 

In the initial data analysis, we found that the scores of the creativity (fluency, flexibility and originality) were not normally distributed, So we corrected the data sets to follow normal distribution, respectively. Then, we conducted two data analyses with the original data sets as the dependent variables and the corrected data sets as the dependent variables. It was found that the results with the original data sets as the dependent variables were similar to those with the corrected data sets as the dependent variables (The results of data analysis using corrected data as the dependent variable are listed in file named Response to Reviewers. Table C1, C2). Since the original scores of creativity are generally used in the existing literature[1, 2], we reported the results of using the original scores as the dependent variables in the original manuscript, but did not report the results of using normalized corrected data sets.

[1] Zabelina D L , Colzato L , Beeman M , et al. Dopamine and the Creative Mind: Individual Differences in Creativity Are Predicted by Interactions between Dopamine Genes DAT and COMT. PLOS ONE, 2016, 11(1).

[2] Runco MA, Noble EP, Reiter-Palmon R, Acar S, Ritchie T, Yurkovich JM. The Genetic Basis of Creativity and Ideational Fluency. Creativity Research Journal. 2011;23(4):376-380. doi: 10.1080/10400419.2011.621859. PubMed PMID: WOS:000299566100010.

Special thanks to you for your good comments.

Reviewer #2:

Major concerns:

1. Response to comment: The overall structure of Introduction is slightly confusing and there is a lack of clarity. For example, the part subtitled “TPH1 rs623580 and creativity” in Introduction was actually about the relation between paternal indifference & neglect in early life and creativity in youth may be moderated by genetic variants.

Reply: Thank you for your helpful advice. We have revised the structure of Introduction. We have added a paragraph subtitled “Role of paternal indifference & neglect” (Page 6, Line 94) to specify the reasons for our study of the relationship between parental indifference & neglect and creativity. Secondly, we have added three paragraphs to specify the reasons for our study examining the paternal indifference & neglect × offspring gender interaction(Page 8, Line 152), the TPH1 rs623580 × offspring gender interaction (Page 9, Line 165), and the three-way interaction(Page 9, Line 174).

2. Response to comment: It’s better going deeper into Discussion. It is not clear what these findings mean for our further understanding of the development of creativity as well as the cultivation of creativity. It should be discussed further.

Reply: Thank you for your helpful advice. We have revised it according to your comments. We have added a paragraph to discuss findings in present study mean for further understanding of the cultivation of creativity (Page 23, Line 424).

3. Response to comment: Both “moderation” and “interaction effects” are very much similar to each other. Mathematically, they both can be modelled by using product term in the regression equation. Researchers often use the two terms as synonyms but there is a thin line between interaction and moderation. When we say X and Z interact in their effects on an outcome variable Y, and there is no real distinction between the role of X and the role of Z. They are both considered predictor variables. Then we identify this effect as interaction effect. While, in case we have clear distinction between the predictor and moderator variables (on the basis of theory) and we are interested to see the impact of predictor on response (affected by moderator), then this effect is known as moderation effect. The authors should carefully choose the term which is more suitable to answer their research question.

Reply: Thank you for your helpful advice. According to the hypothesis of this study, Paternal Indifference and neglect is the predictor variable, and TPH1 genotype and Offspring's Gender are both moderator variables. Thus, in the revised Introduction and Discussion section, we have consistently modified the interaction to a moderation effect according to our hypothesis. Since we used the research method of G×E INTERACTION, we still retained the interaction in the results section.

4. Response to comment: The Results reported the findings of the interaction effect of paternal indifference & neglect and TPH1 rs623580 as well as the interaction effect of TPH1 rs623580 and offspring’s gender on each outcome variable (fluency, flexibility, and originality), but the interaction effect of paternal indifference & neglect and TPH1 rs623580 on fluency, flexibility, and originality didn’t be introduced in detail in Abstract but stated simply “creativity”. Moreover, if there was a different pattern of results for each outcome variable (fluency, flexibility, and originality), it had better be discussed properly in Discussion.

Reply: Thank you for your helpful advice. we have revised the Abstract section according to your comments, and we have added the interaction effect of paternal indifference & neglect and TPH1 rs623580 on fluency, flexibility, and originality. Because the results for each outcome variable (fluency, flexibility, and originality) present the same pattern, thus we did not discuss them separately in the discussion.

Response to comment: Related to this is another issue: in METHODS, the authors did not introduce how they obtained creativity score, although they explained how they obtained fluency, flexibility, and originality scores.

Reply: Thank you for your helpful advice. We are sorry that this part was not clear in the original manuscript. According to Guilford (1967), creativity refers to embodiment of a thought consisting of three characteristics: fluency, flexibility, and originality. Thus, we used these three characteristics as outcome variable in this study, and we did not obtain creativity score.

5. Response to comment: Previous studies indicated that gender difference might be attributed to the interaction effect of genetic and environmental factors on creativity. What is the relationship between the findings in the present study and in previous studies? The cause of non-significant three-way interaction effect on three outcome variables was not discussed in Discussion.

Reply: Thank you for your helpful advice. We have added a paragraph to discuss the non-significant three-way interaction effect on three outcome variables (Page 23, Line 424) in our revised manuscript. In this paragraph, we also have tried to discuss relationship between the findings in the present study and in previous studies.

Minor concerns：

1. Response to comment: The necessary information on Measures in METHODS should be included. For example, “Originality score was calculated by counting the number of unusual ideas given by less the 5% of the sample.” Please described in much more detail about “the sample”. Similarly, “To score flexibility, a category list was first generated for each item, and the flexibility score was the number of different categories used in one participant’s ideas.” Please described in much more detail about “a category list”.

Reply: Thank you for your helpful advice. We have revised Methods section to clarify the criteria by which we evaluate originality scores, and We have added some examples to describe the “a category list” we used to evaluate flexibility scores

2. Response to comment: It is noted that the manuscript needs careful editing. needs to be. Some sentences are awkward.

For example:

Syntax errors：“One possible explanation is that the influence of parental indifference & neglect to children may be differ between mother and father.”

“It is postulated that paternal indifference & neglect in early life would be negatively predict creativity in youth.”

“The findings of the present study contribute to further understanding the role of genetic factors in the pathways that how the early life family environment shapes creativity in adulthood.”

Reply: Thank you for your helpful advice, and we sincerely apologize for the Syntax errors in our previous manuscript. We have thoroughly and completely revised the entire manuscript. And we have made careful amendments to the questions raised in the manuscript.

These comments are quite helpful, and I revised my paper point-by-point. Thank you and the review again for your help!

---

## [Decision Letter · Decision Letter 1]

30 Mar 2020

PONE-D-19-21789R1

Paternal Indifference and neglect in early life and Creativity: Exploring the Moderating Role of TPH1 genotype and Offspring Gender

PLOS ONE

Dear Dr Zhang,

Thank you for submitting your manuscript to PLOS ONE. After careful consideration, we feel that it has merit but does not fully meet PLOS ONE’s publication criteria as it currently stands. Therefore, we invite you to submit a revised version of the manuscript that addresses the points raised during the review process.

We would appreciate receiving your revised manuscript by May 14 2020 11:59PM. To enhance the reproducibility of your results, we recommend that if applicable you deposit your laboratory protocols in protocols.io, where a protocol can be assigned its own identifier (DOI) such that it can be cited independently in the future. For instructions see: http://journals.plos.org/plosone/s/submission-guidelines#loc-laboratory-protocols

We look forward to receiving your revised manuscript.

Kind regards,

Dongtao Wei

Academic Editor

PLOS ONE

Reviewers' comments:

Reviewer's Responses to Questions

**Comments to the Author**

1. If the authors have adequately addressed your comments raised in a previous round of review and you feel that this manuscript is now acceptable for publication, you may indicate that here to bypass the “Comments to the Author” section, enter your conflict of interest statement in the “Confidential to Editor” section, and submit your "Accept" recommendation.

Reviewer #1: All comments have been addressed

Reviewer #2: (No Response)

2. Is the manuscript technically sound, and do the data support the conclusions?

Reviewer #1: Yes

Reviewer #2: Yes

3. Has the statistical analysis been performed appropriately and rigorously? 

Reviewer #1: Yes

Reviewer #2: Yes

4. Have the authors made all data underlying the findings in their manuscript fully available?

Reviewer #1: Yes

Reviewer #2: Yes

5. Is the manuscript presented in an intelligible fashion and written in standard English?

Reviewer #1: Yes

Reviewer #2: Yes

6. Review Comments to the Author

Reviewer #1: (No Response)

Reviewer #2: 1. The manuscript improved and some parts were clarified. The revised version is more well-organized than the previous one. However, the statement “Findings from the current study suggested that the A allele of TPH1 (rs623580) might be a risk allele for creativity, and the long-term negative influence of paternal indifference & neglect in early life on individuals’ creativity in youth depending on TPH1 genotype.” in Abstract is not prudent. One should be cautious in drawing general conclusions from the specific results.

2. I suggest the authors add the accurate references to the Divergent Thinking Test scoring methods for obtaining fluency, flexibility, and originality scores. Moreover, the scoring methods was still not described sufficiently. Given the relatively low Cronbach’s alpha for flexibility, more detailed information about the scoring methods should be provided.

3. Please double check the tense issue in Introduction and Discussion.

7. PLOS authors have the option to publish the peer review history of their article (what does this mean?). If published, this will include your full peer review and any attached files.

Reviewer #1: Yes: Li Wenfu

Reviewer #2: No

---

## [Author Response · Author response to Decision Letter 1]

7 May 2020

Responds to the reviewers’ comments: 

Reviewer #2: 

1. Response to comment: The manuscript improved and some parts were clarified. The revised version is more well-organized than the previous one. However, the statement “Findings from the current study suggested that the A allele of TPH1 (rs623580) might be a risk allele for creativity, and the long-term negative influence of paternal indifference & neglect in early life on individuals’ creativity in youth depending on TPH1 genotype.” in Abstract is not prudent. One should be cautious in drawing general conclusions from the specific results.

Reply: Thank you for your helpful advice. We have rewritten the last sentence of our Abstract section according to your comments (Page 2, Line 23). We summarized the results of this study, and tried to avoid drawing inappropriate general conclusions.

2. Response to comment: I suggest the authors add the accurate references to the Divergent Thinking Test scoring methods for obtaining fluency, flexibility, and originality scores. Moreover, the scoring methods was still not described sufficiently. Given the relatively low Cronbach’s alpha for flexibility, more detailed information about the scoring methods should be provided.

Reply: Thank you for your helpful advice. We have revised Methods section according to your comments. Fristly, we added accurate references to the Divergent Thinking Test scoring methods for obtaining fluency, flexibility, and originality scores(Page 12, Line 225, Line 235). Secondly, we revised sentence whcih described the method for obtaining originality scores to further clarify the criteria we used (Page 12, Line 227). Moreover, we added sentences to describe the process we used to evaluate flexibility scores in detail, and we also added examples to describe the “a category list” (Page 12, Line 228). 

3. Response to comment: Please double check the tense issue in Introduction and Discussion.

Reply: Thank you for your helpful advice. We have thoroughly and completely revised the entire manuscript. And we have made careful amendments to the questions raised in the manuscript.

We tried our best to improve the manuscript and made some changes in the manuscript. These will not influence the content and framework of the paper. And here we did not list the changes but marked in red in revised paper.

We appreciate for reviewer’s warm work earnestly, and hope that the correction will meet with approval.

---

## [Editor Report · Decision Letter 2]

18 May 2020

Paternal Indifference and neglect in early life and Creativity: Exploring the Moderating Role of TPH1 genotype and Offspring Gender

PONE-D-19-21789R2

Dear Dr. Zhang,

We are pleased to inform you that your manuscript has been judged scientifically suitable for publication and will be formally accepted for publication once it complies with all outstanding technical requirements.

With kind regards,

Dongtao Wei

Academic Editor

PLOS ONE
---

## [Editor Report · Acceptance letter]

17 Jul 2020

PONE-D-19-21789R2 

Paternal Indifference and neglect in early life and Creativity: Exploring the Moderating Role of *TPH1* genotype and Offspring Gender 

Dear Dr. Zhang:

I'm pleased to inform you that your manuscript has been deemed suitable for publication in PLOS ONE. Congratulations! Your manuscript is now with our production department. 

Kind regards, 

on behalf of

Dr. Dongtao Wei 

Academic Editor

PLOS ONE